# Experimental Study on Water Electrolysis Using Cellulose Nanofluid

**Dongnyeok Choi and Kwon-Yeong Lee \*** 

Handong Global University, 558 Handong-Ro, Heunghae-eup, Buk-gu, Pohang-si, Gyeongbuk-do 37554, Korea; cdn4400@hanmail.net

**\*** Correspondence: kylee@handong.edu; Tel.: +82-54-260-1176

**Abstract:** Hydrogen energy is considered to be a future energy source due to its higher energy density as compared to renewable energy and ease of storage and transport. Water electrolysis is one of the most basic methods for producing hydrogen. KOH and NaOH, which are currently used as electrolytes for water electrolysis, have strong alkalinity. So, it cause metal corrosion and can be serious damage when it is exposed to human body. Hence, experiments using cellulose nanofluid (CNF, $C_6H_{10}O_5$) as an electrolyte were carried out to overcome the disadvantages of existing electrolytes and increase the efficiency of hydrogen production. The variables of the experiment were CNF concentration, anode material, voltage applied to the electrode, and initial temperature of the electrolyte. The conditions showing the optimal hydrogen production efficiency (99.4%) within the set variables range were found. CNF, which is not corrosive and has high safety, can be used for electrolysis for a long period of time because it does not coagulate and settle over a long period of time unlike other inorganic nanofluids. In addition, it shows high hydrogen production efficiency. So, it is expected to be used as a next-generation water electrolysis electrolyte.

**Keywords:** water electrolysis; electrolyte; cellulose nanofiber; nanofluid

---

## 1. Introduction

Due to the recent increase in economic growth, population growth, mass production, and mass consumption, fossil fuels are rapidly depleting and there is a rise in environmental pollution. Additionally, nuclear energy, which accounts for a large part of the current electricity generation, is highly dangerous considering power plant accidents, such as the Fukushima nuclear power plant accident. Therefore, it is important to develop eco-friendly renewable energy, and investment and research on energy sources such as solar power, geothermal power, and wind power is actively being carried out. However, the above-mentioned renewable energy sources are limited due to low energy density and unstable energy production. This threatens the stability of the power network and reduces the power generation efficiency. Hence, mass power production and storage technology using hydrogen is gaining attention.

Hydrogen energy has three times the energy density of natural gas and gasoline [1], a very high conversion efficiency to electric power, is relatively easy to store and transport [2], has a purity of 99.9% and does not need further purification [3], and does not emit greenhouse gas when used as energy. Thus, it can replace existing fossil fuels as an energy source and an energy storage medium. There is also a need for efficient hydrogen production technology as well as the development of hydrogen storage devices. However, the corresponding economical and environment-friendly methods have not yet been developed and various studies are being conducted across the globe for the same.

Currently, water electrolysis is the simplest method of producing hydrogen, but accounts for only 4% of all the hydrogen production methods [4,5]. This is because the technology is limited due to high costs associated with a higher power consumption relative to production, high installation cost, equipment maintenance cost, and low equipment durability and safety [6,7]. The durability and safety issues of electrolysis are mostly caused by electrolytes such as KOH and NAOH, which are commonly used in water electrolysis. These electrolytes are corrosive towards metals due to their strong alkalinity. This damages the electrodes, and therefore, requires frequent replacement of the electrodes. Additionally, there are limitations during operations, such as sealing of the container and frequent ventilation when storing and operating the solution, as the electrolytes can cause serious damage when exposed to the skin or eyes and can affect the respiratory system when exposed to air. In case of KOH, there is a risk of explosion due to flammability and requires special care during storage and handling. Recently, various studies have been conducted to overcome the limitations in hydrogen production and to increase its efficiency through water electrolysis. De Souza et al. [8] used imidazolium ionic as an electrolyte for hydrogen production through water electrolysis, found that the electrolyte improved efficiencies of hydrogen production. Nagai et al. [9] studied the effect of the distance between electrodes on hydrogen generation efficiency. Wang et al. [10] conducted a study to improve hydrogen production through water electrolysis using an ultra-gravity field. Mandal et al. [11] experimented and analyzed the changes in the amount of hydrogen produced according to the concentration during electrolysis. In this study, Cellulose Nanofibers (CNF) were used as electrolyte in the electrolysis of water in order to raise the efficiency of hydrogen production, reduce the production cost, and overcome the problems of using alkaline electrolytes.

CNF, which is called a "dream material", is an organic compound that forms the basic structure of plant cell walls. It is composed of the hydroxyl structure with hydrogen and oxygen and is weakly alkaline when it comes into contact with water. It is stronger than iron, is lightweight, and does not expand even when exposed to heat. Also, it can be used for electrolysis for a long period of time because it does not coagulate and settle over a long period of time unlike other inorganic nanofluids. Hence, it can be applied to various fields, such as automobile parts, glass substitutes, and biomedical materials.

## 2. Materials and Methods

### 2.1. CNF Production

CNF does not have electrical conductivity. However, when CNF is fabricated through the method described by Saito et al. [12], the carboxyl group is replaced with $C_6OH$ and becomes charged. Briefly, a coffee filter was used as a raw material for cellulose and oxidized using three catalysts: 2,2,6,6-Tetramethylpiperidin-1-yl-oxyl (TEMPO), NaBr, and NaClO. This is a useful surface modifying method for fabricating nanofibers without significantly degrading the crystallinity of the cellulose. The surface modified cellulose can be obtained in the form of nanofibers dispersed in water using a simple mechanical treatment. After diluting the CNF gel in water according to the test concentration, ultrasonic dispersion was performed for 1 h using a sonicator to disperse it in water. Therefore, when CNF is completely dispersed in water, it acts as an electrolyte. Each fiber has 3–4 nm of width and 2–3 μm of length [13]. CNF is negatively charged, so zeta-potential indicates the stability of CNF. In the experiment, we used CNF of which pH was above 5, and the average value of zeta-potential was −53.3 mV (usually considered stable when the absolute value is more than 30 mV). CNF was very stable because electrostatic force took the fiber apart, and the stability of the mixed liquid is also high because of its very low reactivity with water and very low corrosivity. Also, since it is an eco-friendly raw material extracted from plants, it is harmless to the human body, easy to dispose of, and economical due to easy resource acquisition. Additionally, Hwang et al. [13] found that the coagulation and precipitation of nanofibers did not occur even when CNF was operated for a long time.

### 2.2. Experimental Apparatus and Method

For the electrolysis of the mixture of water and CNF, the Hofmann electrolysis apparatus shown in Figure 1 was used. It is made of acrylic for visualization of the gas. When oxygen and hydrogen collect in the anode and the cathode, respectively, the gas pushes out the water. The capacity was 200 mL, the distance between each electrode was 3.2 cm, and the effective surface area of the electrode was 5.62 cm$^2$. The amount of gas that could be collected was 18 mL for both oxygen and hydrogen.

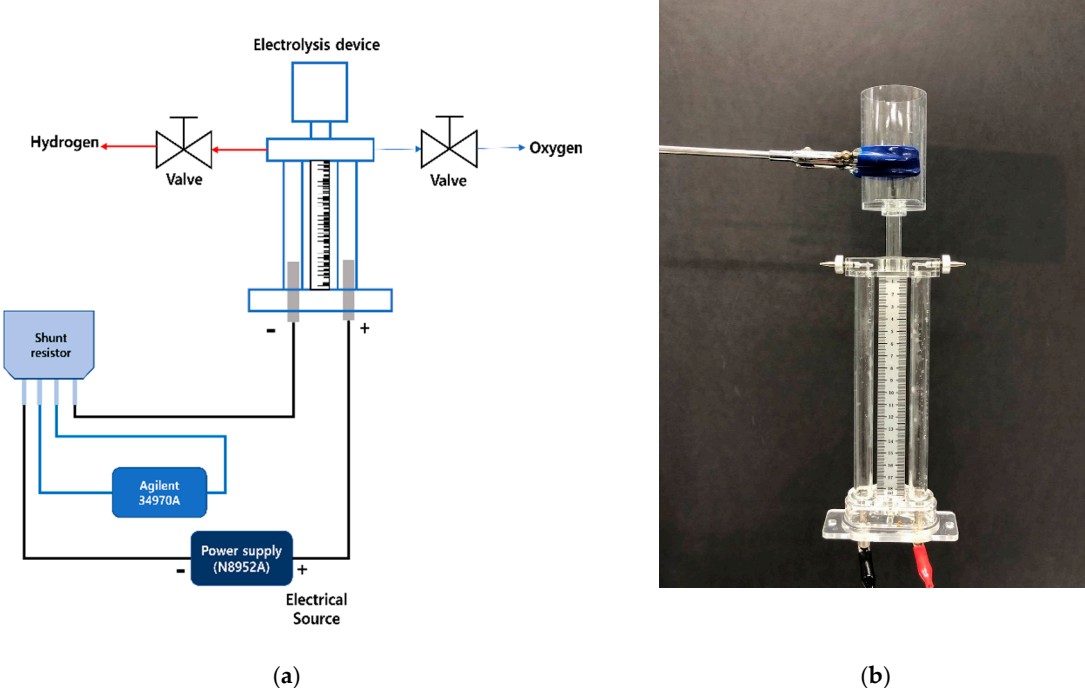

(**a**)                                                                                                                                              (**b**)

**Figure 1.** Hofmann electrolysis apparatus (**a**) schematic (**b**) picture.

The hydrogen production efficiency was obtained by experimenting with the following four variables: the concentration (wt %) of the electrolyte, the material of the electrode, the voltage applied to the electrode, and the initial temperature of the electrolyte. The optimum hydrogen production efficiency condition was found by comparing the hydrogen generation efficiency between the variables. The hydrogen generation efficiency under these conditions was also compared to the hydrogen generation efficiency when using NaOH as the electrolyte.

Experiments were conducted for 75 min. A power supply (N8952A, Keysight) with accuracy of ± 0.0015% was used. It can supply voltage and current up to 210 A, 200 V, respectively. It can also measure voltage with accuracy of ± 0.0015%. Both the electrolyte temperature and the atmosphere temperature were measured using thermocouples temperature, and the voltage was measured using DMM. During the electrolysis experiment, data on the variations in temperature and current with time were collected using a data acquisition system (34,770 A, Keysight) at 10 s intervals. The gas produced at the cathode and the anode was measured via a 0.1 mL spacing scale. Chakik et al. [14] summarized Equations (1)–(3) with respect to the electrolysis efficiency.

$$V_{H2Ideal} = \frac{I * V_M * t}{2 * F} \tag{1}$$

$V_{H2Ideal}$ is the ideal hydrogen generation volume, $I$ is the current [A], $V_M$ is the Molar volume in ideal state (24.47 L/mol), $t$ is time [s], and $F$ is the Faraday constant (96,485 s A/mol).

$$V_{H2Real} = V_{H2Measured} * \frac{T_{standard}}{T_{measured}} \tag{2}$$

$V_{H_{2Real}}$ is the volume of the actual hydrogen generated, $V_{H_{2Measured}}$ is the hydrogen volume obtained from the experiment, and $T_{standard}$ and $T_{measured}$ are reference temperature (298.15 K) and the average temperature [K], respectively, measured during electrolysis.

$$\text{Efficiency (\%)} = \frac{V_{H_{2Real}}}{V_{H_{2Ideal}}} * 100 = \frac{V_{H_{2Measured}} * \frac{T_{standard}}{T_{measured}}}{\frac{I*V_M*t}{2*F}} \tag{3}$$

Equation (3) shows the formula for obtaining the hydrogen generation efficiency, which can be expressed as the ratio of the hydrogen volume actually generated to the ideal volume. In order to obtain the current of the entire electrolysis system, a 1.0 Ω shunt resistor was connected between the cathode and the power supply. After the electrolysis began, the voltage applied to the shunt resistor was measured at intervals of 10 s using the data acquisition system and the current was calculated by dividing the measured voltage by the shunt resistance value. The average current value was calculated by dividing the sum of the current data obtained during the experiment by the number of data points. The pH of the electrolyte was also measured before and after the experiment using a pH meter.

## 3. Results and Discussion

### 3.1. Experiment According to Electrolyte Concentration

The electrolyte concentration was varied between 0.5 wt %, 1.0 wt %, 1.5 wt %, 1.8 wt %, and 2.2 wt % under the voltage of 10 V, an initial temperature of the electrolyte of 26 °C, and the platinum anode SUS304 cathode. The results obtained are shown in Table 1. As shown in Figure 2, the higher the concentration of the electrolyte, the higher is the current value. In addition, the highest hydrogen generation efficiency of 89% was obtained when the concentration was 1.8 wt %, as more ions were released in the aqueous medium with an increase in the concentration of the electrolyte. When the concentration of the electrolyte was 2.2 wt %, the generated gas did not rise because CNF was coated on the anode, and the generated oxygen was trapped due to the CNF coating, as shown in Figure 3. However, except in the case of the concentration of 2.2 wt %, it seems that the generated gas was able to rise at the anode as the rising force was greater than the gas holding force exerted by the CNF coated on the anode. Figure 4 shows the change in pH measured with a pH meter before and after the experiment with respect to the electrolyte concentration. The concentration of electrolyte and the number of ions are proportional and the change in pH with respect to the number of ions shows a '-log scale'. This indicates that the smaller the number of ions, the greater is the change in pH. The change in pH decreased very slowly from 0.5 to 1.5 wt %. However, at 1.8 wt %, the pH change was greatly reduced. It is believed that the pH change was not large even after electrolysis because there were much more ions contained at a concentration of 1.8 wt % than the amount of ions required for electrolysis. Therefore, when the concentration of the electrolyte was 1.8 wt %, it resulted in not only the best hydrogen generation efficiency but also a near neutral pH after the experiment due to the minor change in pH. It resulted in the high stability and the small causticity of the solution. After all the experiments in this study, agglomeration and sedimentation of CNF were not discovered.

**Table 1.** Experimental results according to the concentration of CNF.

| Concentration wt % | Hydrogen Volume mL | Average Temperature [K] | Average Current [mA] | Efficiency [%] |
|---|---|---|---|---|
| 0.5 | 5.0 | 295.4 | 11.6 | 76.3 |
| 1.0 | 9.2 | 294.5 | 19.8 | 82.5 |
| 1.5 | 12.8 | 298.4 | 27.0 | 83.1 |
| 1.8 | 15.0 | 293.8 | 30.0 | 89 |

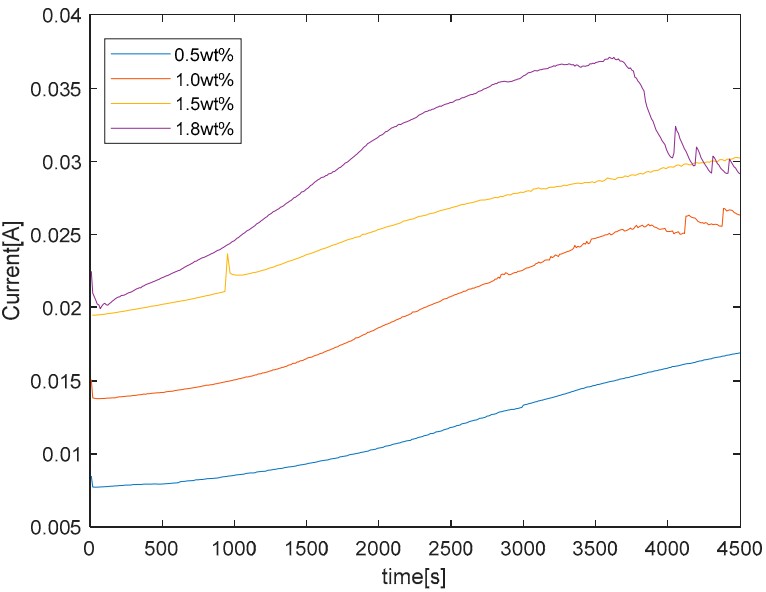

**Figure 2.** Current according to the electrolyte concentration.

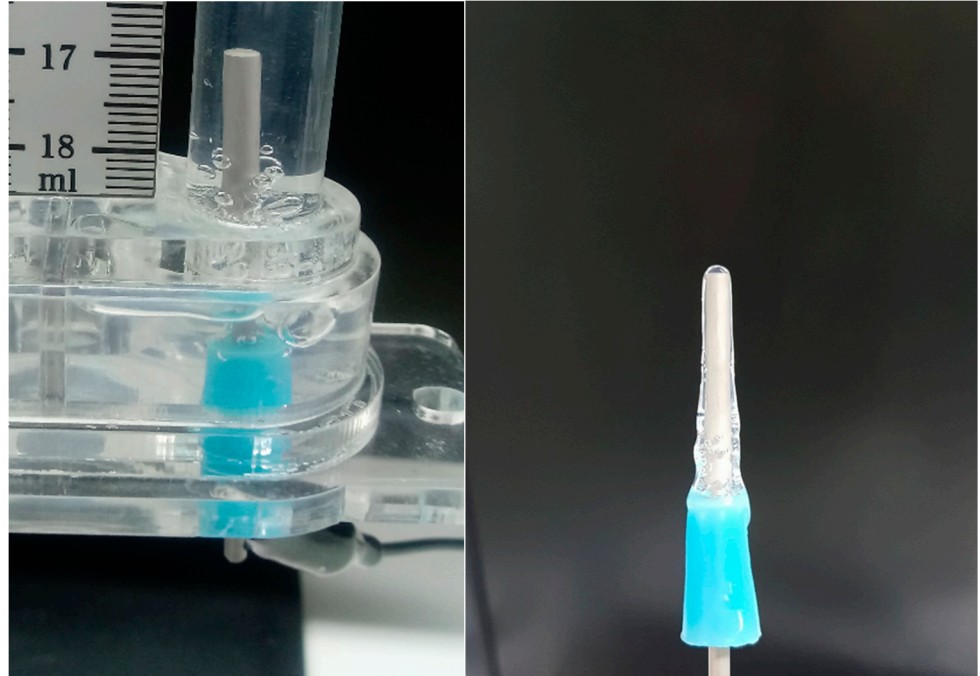

**Figure 3.** Coated CNF at the anode.

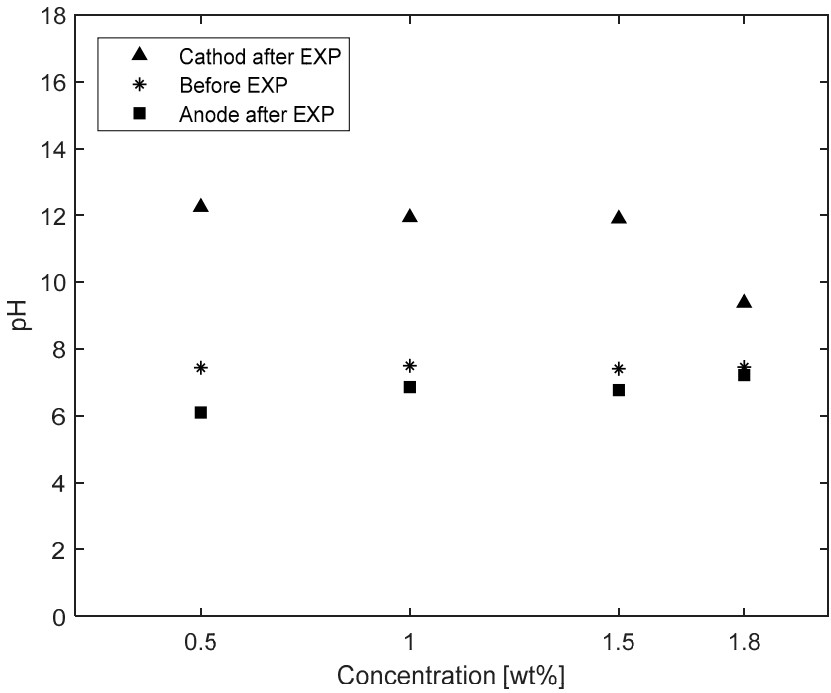

**Figure 4.** pH change of the electrode before and after the experiment.

### 3.2. Comparison of the CNF Solution Sample Replacement

Because of the shortage of the first CNF sample solution, another sample CNF solution was used from Experiment 3.2. In order to confirm the effect of the change of sample on the electrolysis efficiency, the experimental results were compared by varying the samples under a voltage of 10 V, an initial temperature of the electrolyte of 26 °C, and the platinum anode and SUS304 cathode.

The hydrogen produced in the previous CNF sample experiments was 2.5 times more than that in the new CNF samples, as shown in Table 2. However, as the amount of hydrogen generated decreased, the current also reduced, as shown in Figure 5, and the hydrogen generation efficiency error between the samples was not great, less than 6%.

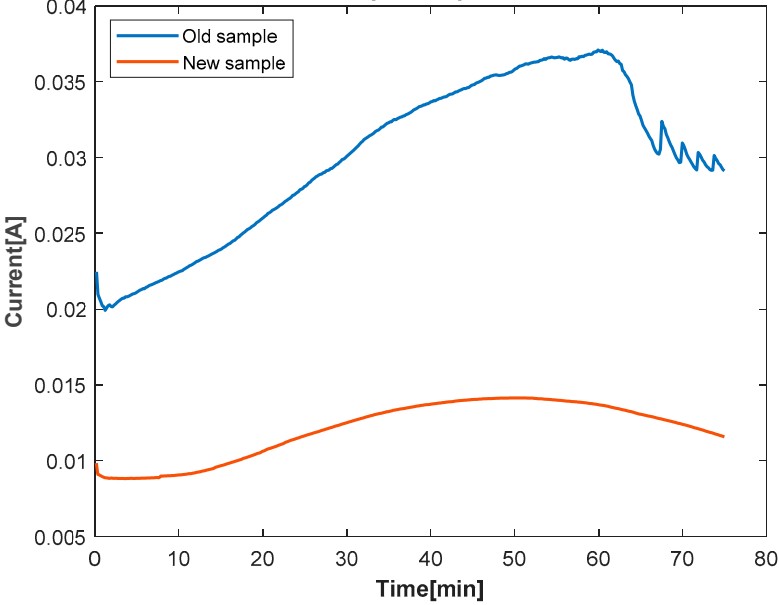

**Figure 5.** Current according to the CNF solution sample.

**Table 2.** Comparison of CNF solution sample results.

| Sample | Hydrogen Volume [mL] | Average Temperature [K] | Average Current [mA] | Efficiency [%] | Error [%] |
|--------|---------------------|------------------------|---------------------|----------------|-----------|
| Old | 15.0 | 293.8 | 30.0 | 89 | 5.7 |
| New | 5.8 | 298.4 | 12.1 | 83.8 | |

## 3.3. Experiment According to Anode Material

Under the voltage of 10 V, an initial temperature of the electrolyte of 26 °C, an electrolyte concentration of 1.8 wt %, and SUS304 cathode, SUS304 and platinum were alternately applied as anode materials and the experiment was conducted as shown Table 3 and Figure 6. When SUS304 was used as an anode, the hydrogen generation efficiency was 88%, higher than that of using platinum.

**Table 3.** Experimental result according to anode material.

| Anode Material | Hydrogen Volume [mL] | Average Temperature [K] | Average Current [mA] | Efficiency [%] |
|----------------|---------------------|------------------------|---------------------|----------------|
| Stainless | 8.0 | 298.7 | 15.9 | 88.0 |
| Platinum | 5.8 | 298.4 | 12.1 | 83.9 |

Figure 6a,b show that when SUS is used as an anode material, current flow and hydrogen generation is higher than when platinum is used. As per Figure 6a, immediately after the voltage is applied, both the SUS304 case and the platinum case show that the current value rapidly decreased at first and then increased. This was due to the initial voltage barrier. In the case of SUS304, since the initial voltage barrier was small as compared to platinum, the rising rate of the current value was faster. In addition, the current graph rose with time and then dropped down. This was because the ionic activity caused the ions to collide with each other, thereby decreasing the current flow. After the experiment, agglomeration and sedimentation of CNF were not discovered.

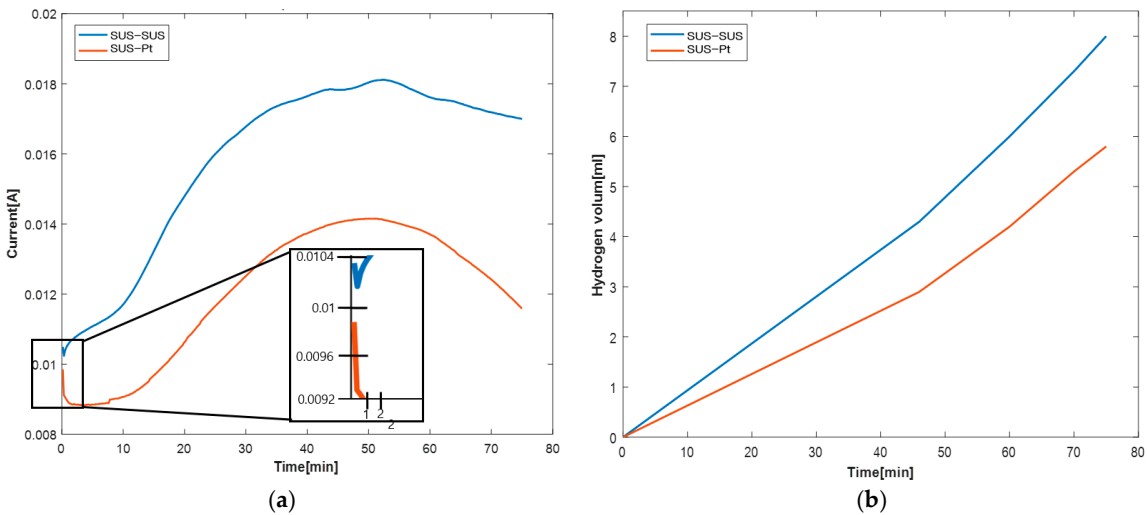

**Figure 6.** Experimental result according to anode material (**a**) Current (**b**) Hydrogen generation.

## 3.4. Experiment According to Applied Voltage

Table 4 shows the results obtained by applying voltages of 3 V, 6 V, and 10 V to the electrolysis apparatus under an initial temperature of the electrolyte of 26 °C, an electrolyte concentration of 1.8 wt %, and the SUS304 anode cathode. It should be noted that the electrolysis apparatus used until the experiment 3.3 was defective and a new electrolysis apparatus was used from the experiment 3.4.

The results of the old device at experiment 3.3 with the voltage 10 V and the new device at experiment 3.4 with the voltage 10 V were slightly different with efficiencies of 88.01% and 92.61%, respectively, showing 4.6% of error. As shown in Table 4, it was found that the current and the amount of hydrogen generated increased as the applied voltage increased, and the highest hydrogen generation efficiency was obtained when 10 V was applied. As shown in Figure 7a, the current increased with the increase in voltage due to ion activity in the solution. On the other hand, as the voltage was lowered, the current change was negligible. In Figure 7b, it can be seen that the higher the applied voltage, the larger was the slope of the hydrogen volume occurring over time. In other words, the current change according to the voltage applied to the electrolytic apparatus and the activity of the ions in the solution are proportional.

**Table 4.** Experimental results according to the applied voltage.

| Voltage [V] | Hydrogen Volume [mL] | Average Temperature [K] | Average Current [mA] | Efficiency [%] |
|---|---|---|---|---|
| 3 | 1.1 | 298.1 | 2.5 | 77.1 |
| 6 | 4.0 | 297.0 | 7.9 | 89.1 |
| 10 | 10.8 | 300.1 | 20.3 | 92.6 |

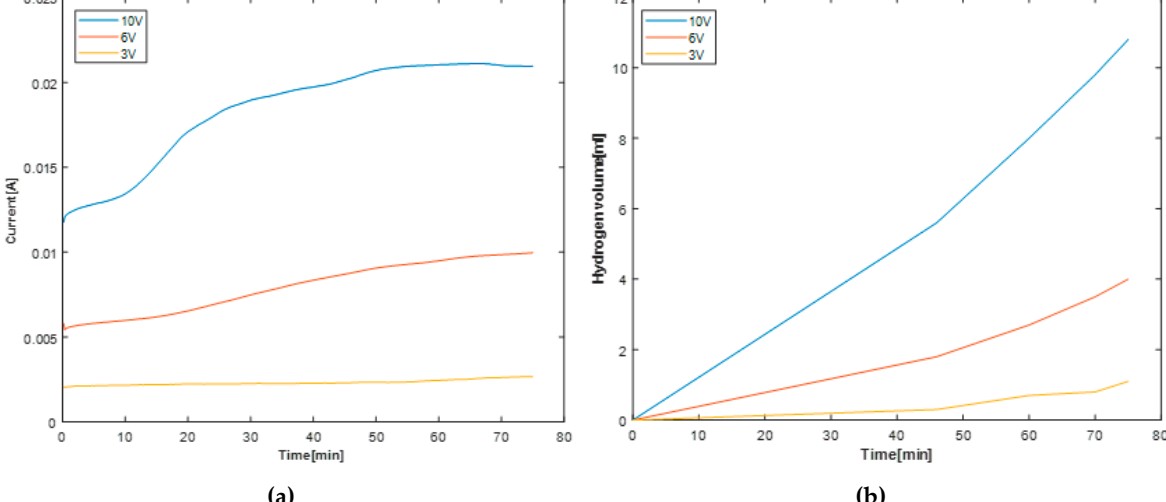

(a)         (b)

**Figure 7.** Experimental results according to the applied voltage (**a**) Current (**b**) Hydrogen generation.

Hydrogen production efficiency depends not only on the number of ions present in the electrolyte solution but also on the effect of mobility in the solution [11]. Therefore, the number and mobility of ions must be taken into account in the current change that directly affects the hydrogen production efficiency. First, as the magnitude of the applied voltage increased, the magnitude of the lost overvoltage also increased with time. As the temperature increased gradually due to the lost overvoltage, the ionic activity in the solution and the current increased. However, since the ion mobility was slightly reduced by the highly active ions, a high voltage was not necessarily suitable for the electrolysis. In addition, since the current density was reduced due to the reduced effective surface area as a result of the adhesion of the bubbles on the electrode surface, the applied voltage range should be considered in terms of hydrogen generation efficiency [14]. In the case of voltage experiments at 3 V, 6 V, and 10 V, as the voltage increases, the magnitude of the current and the amount of generated hydrogen increase, so it is confirmed that the influence of the reduction of the current density due to the interruption of the ion mobility and the reduction of the effective surface area is insignificant. Therefore, it is reasonable to carry out the electrolysis experiments at the voltage of 10 V, which showed the highest average current of 0.0203 mA and the hydrogen generation efficiency of 92.6%.

*3.5. Experiment According to Initial Temperature of Electrolyte*

　　Under the electrolyte concentration of 1.8 wt %, voltage of 10 V, and SUS304 anode and cathode, which are the optimal hydrogen generation efficiency conditions derived from previous experiments, the initial temperature of the electrolyte was set at 52 °C, 26 °C and 9 °C, and the effect of the initial temperature on the electrolysis was examined. Since the Hofmann electrolytic apparatus was made of acrylic and deformed at a temperature of 80 °C or higher, the highest initial temperature was set at 52 °C, and the electrolyte was placed in a Pyrex vessel and heated using a hot plate. The lowest temperature of 9 °C, which is the lowest temperature that can be cooled through cooling water, was set as the lowest temperature of the experiment. During the experiment, since it was difficult to maintain the temperature of the electrolyte due to the characteristics of acrylic, a heat loss was naturally induced. Figure 8 shows the change in temperature of the electrolyte during the experiment. As the experiment progressed, the temperature of the electrolyte reached closer to room temperature (22–26 °C). Table 5 and Figure 9 show that the average current and the amount of hydrogen generated were higher as the initial temperature of the electrolyte increased. In the case of highest initial temperature of the electrolyte, the average current was 22.1 mA and the hydrogen generation efficiency was the highest at 99.4%.

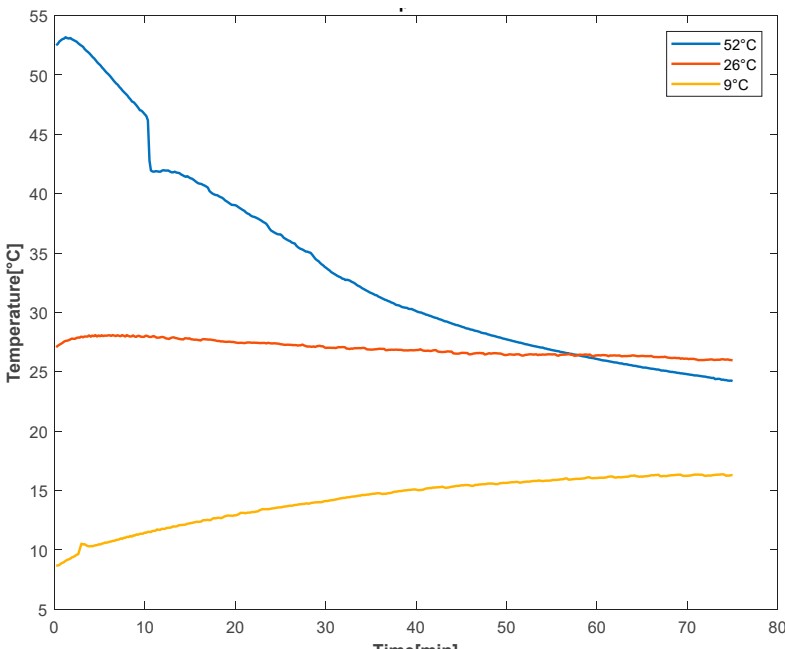

**Figure 8.** Electrolyte temperature change.

**Table 5.** Experimental results according to initial temperature of the electrolyte.

| Initial Electrolyte Temperature [°C] | Hydrogen Volume [mL] | Average Temperature [K] | Average Current [mA] | Efficiency [%] |
|:---:|:---:|:---:|:---:|:---:|
| 9 | 9.3 | 297.6 | 16.7 | 97.8 |
| 26 | 10.8 | 300.1 | 20.3 | 92.6 |
| 52 | 12.7 | 302.1 | 22.1 | 99.4 |

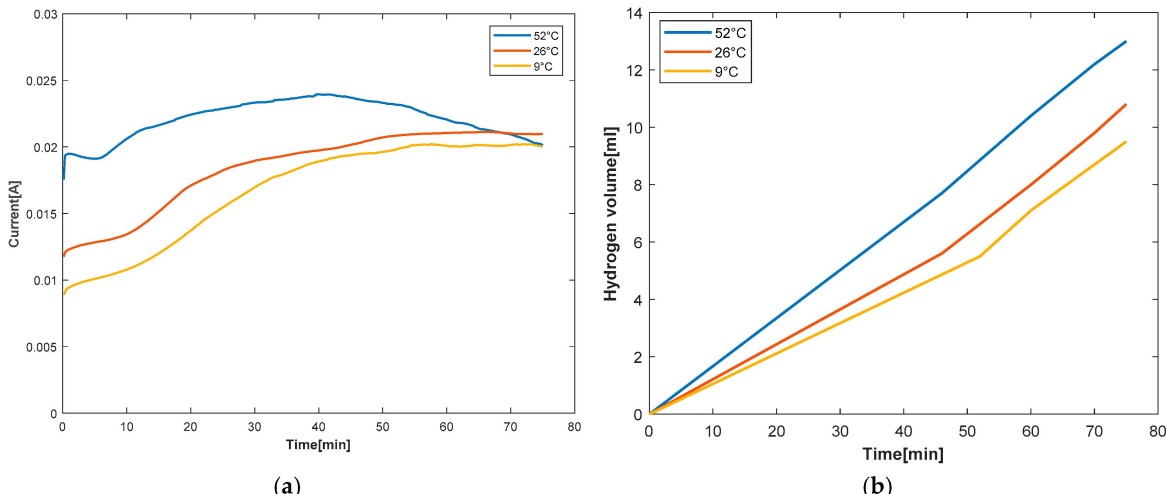

(**a**)                                                        (**b**)

**Figure 9.** Experimental results according to initial temperature of electrolyte (**a**) Current (**b**) Hydrogen generation.

It can be seen that the higher the initial temperature of the electrolyte, the higher was the average current value including the amount of hydrogen generated. Increasing the operating temperature is a common method to improve the electrolysis efficiency because as the temperature of the electrolyte increases, the transfer coefficient and the Tafel slope, which shows the rate of hydrogen generation, decreases [15]. Moreover, additional energy by heat is increased and the required Gibb's free energy is reduced [16]. Therefore, as the temperature of the electrolytic solution increased, lower voltage was required for generating hydrogen and the efficiency increased. The higher the temperature of the electrolyte, the higher was the current flow during electrolysis. This is because at higher temperatures of the electrolyte, the ions moved more freely, and the current flow increased. Also, in the latter half of the electrolysis, the current of the electrolyte at a high initial temperature dropped sharply and was similar to that of the electrolyte at a low initial temperature. This is because, as shown in Figure 8, the temperature of the electrolyte approaches the room temperature (22–26 °C) over time. In this experiment, it shows higher efficiency at 9 °C than at 26 °C. This can be explained using the concept of reversible potential. Reversible potential refers to the minimum voltage applied to the two electrodes of the electrolysis cell to cause the electrolysis reaction of water. Oliver et al. reported that the lower the temperature of the solution, the higher the reversible potential [17]. Similarly, in this study, since the reversible potential at the case of initial temperature 9 °C was higher than that of 26 °C, the current flowing at the case of initial temperature 9 °C was lower than that of 26 °C. Therefore, even though more hydrogen was produced at the case of 26 °C, the hydrogen production efficiency was low because the current flowing was large. After the experiment, agglomeration and sedimentation of CNF were not discovered.

## 4. Conclusions

Water electrolysis is the most basic and an important technology towards developing hydrogen energy. Various studies have been carried out for this globally, including research on electrodes and electrolytes. This study was conducted to overcome the issues of using conventional electrolytes by using CNF as an electrolyte for electrolysis of water, to improve the productivity of hydrogen, to find the optimum conditions of generating hydrogen efficiently, and to demonstrate its applicability as an electrolyte. Experiments were carried out using CNF as an electrolyte for electrolysis of water with several variables: electrolyte concentrations were 0.5, 1.0, 1.5, 1.8 wt %, 2.2 wt % and SUS304 and platinum were used as anode materials. The voltages applied to the electrodes were varied between 3 V, 6 V, and 10 V and the initial temperatures of the electrolyte were set to 9 °C, 26 °C, and 52 °C. The highest hydrogen generation efficiency was obtained when the electrolyte concentration was

1.8 wt %, the anode material was SUS304, the applied voltage was 10 V, and the electrolyte initial temperature was 52 °C. The obtained efficiency was 99.4%.

CNF can be a cost-effective option because it is easier to supply the main raw materials compared to KOH and NaOH, which are used in the existing electrolysis. Also, compared to conventional electrolytes, CNF has low reactivity with water and excellent fluid stability due to low corrosiveness which reduces the risk of device corrosion. Also, it can be used for electrolysis for a long period of time because it does not coagulate and settle over a long period of time unlike other inorganic nanofluids. Finally, it will improve the proportion of hydrogen energy used and help in combating the environmental problems of associated with the existing energy sources.

**Author Contributions:** Conceptualization, D.C. and K.-Y.L.; methodology, D.C. and K.-Y.L.; validation, D.C. and K.-Y.L.; formal analysis, D.C. and K.-Y.L.; investigation, D.C. and K.-Y.L.; resources, D.C. and K.-Y.L.; data curation, D.C. and K.-Y.L.; writing—original draft preparation, D.C.; writing—review and editing, K.-Y.L.; visualization, D.C.; supervision, K.-Y.L.; project administration, D.C. and K.-Y.L.; funding acquisition, D.C. and K.-Y.L. All authors have read and agreed to the published version of the manuscript.

**Funding:** This work was supported the Basic Research Program of the National Research Foundation of Korea (NRF) grant funded by the Korean government (MSIT; Ministry of Science and ICT) (No. NRF-2017R1C1B5017640).

**Conflicts of Interest:** The authors declare no conflict of interest. The funders had no role in the design of the study; in the collection, analyses, or interpretation of data; in the writing of the manuscript, or in the decision to publish the results.

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
