# Peer review of "Experimental Study on Water Electrolysis Using Cellulose Nanofluid"

_fluids, doi:10.3390/fluids5040166_

Round 1

Reviewer 1 Report

The authors have experimentally studied the water electrolysis and enhancement of hydrogen production efficiency using cellulose nanofluid. The work seems to be original and the experimental results provide some useful benefits of cellulose nanofluid for water electrolysis. Thus, the manuscript is recommended for publication to Fluids after minor revision. Please the authors address the following questions:

  1. In the abstract and introduction, it was mentioned that the cellulose nanofluid does not undergo agglomeration and sedimentation of nano particles unlike other inorganic nanofluids. Didn't agglomeration and sedimentation occur even after the actual experiment? If so, please mention it on the Result part.

  1. The experimental device was changed once during your study. Was there a difference between the test results of the old and new devices? Please compare the difference between the two results.

  1. Can the authors provide the relevant physical properties of cellulose nanofluid such as density, latent heat and so on?

Reviewer 2 Report

Review of manuscript fluids-930192

Title: Experimental Study on Water Electrolysis Using Cellulose Nanofluid

Authors: Dongnyeok Choi and Kwon-Yeong Lee

Recommendation: Major revision

In this paper, the authors have presented cellulose nanofluid as electrolyte and checked by different parameter for showing the efficiency. Method and material processing along with definition of efficiency are presented. Analysis based on electrolyte-concentration, current, applied voltage and temperature are shown as the justification of electrolytes. They have claimed better performance based on the results. The findings are interesting, however, require more clarification before publishing.

Minor points:

  1. Surface modifying method is ambiguous to understand, the main basis of this research. It is told that TEMPO, NaBr and NaClo are being used as catalysts for fabricating nanofibers and nanofibers are dispersed in water using a simple mechanical treatment. How long did this dispersion process take place and what kind of mechanical treatment was done? How it can be ensured that the neutral catalyst is actually being charged or it never being charged but catalyst are the mixed in electrolyte to show charge?

  1. From table-1, why the avg. temperature is higher for 1.5 wt% than other higher and lower concentrations?

  1. What is the reference and justification of equation 3 which is the base of efficiency? In table-5, initial electrolyte temperature 9oC shows higher efficiency than 26oC, however, 52oC shows highest efficiency, which is overall out of trend. Why such kinds of discrepancy is occurring.

  1. From page 9 it is mentioned that as the voltage increase, the magnitude of current and amount of hydrogen increased. Up to which range of this proportionality of voltage with current and hydrogen can be valid or that is like for unlimited scale?

  1. The CNF sample had been replaced during the experiment and author showed that because of changing the sample 5.68% error has been found. Is this amount of error acceptable? What will be the result if the sample replaced again? What is the guaranty that every time with changing sample, the error will be near 5.68%?

  1. In Fig.5 shows the value of pH for “cathode after EXP” is almost constant for 0.5 to 1.5 electrolyte concentration. After 1.5 its shows drop, what is the reason for this certain drop? The authors should explain the reason clearly.

  1. Authors have use SUS304 as a cathode in this study. What is the reason for this choice?

  1. In Fig.7 (a) and (b), authors have written in legend “SUS-SIS” and “SUS-SUS” is it correct?

  1. Authors should present the magnify view at initial value of Fig.7 (a).

Reviewer 3 Report

The submitted manuscript:

No. fluids-930192

Title : EXPERIMENTAL STUDY ON WATER ELECTROLYSIS USING

CELLULOSE NANOFLUID

Authors: Dongnyeok Choi, Kwon-Yeong Lee

was reviewed.

General comment;

The authors investigated the efficiency of hydrogen production of CNF using a conventional Hofmamm voltameter. The dependence of the production rate of hydrogen on CNF solution concentration, current density, and initial temperature was examined to find an optimal value of production rate. The manuscript is well written and the subject of the study is interesting, however, the manuscript has two fatal flaws in method of analysis and, has many improper descriptions in the manuscript. Firstly, the authors claim that the two different sample solutions were used in the study since the first sample had been consumed before the completion of the series of experiment. However, any descriptions of detail of the difference of material properties are not given. Table.2 and Fig.6 show significant discrepancies between them. The given data up to the section 3.2 should be updated. In particular, Fig.5 may originally be very important, however these ones lose values in this manuscript. Secondary, method of temperature measurement and method of the evaluation of temperature were not described. Was the temperature of the whole parts of the voltameter kept constant prior to the experiment? And what was the room temperature? And how the temperature was measured? All these things are unclear.

Minor comments;

In the line 73, length of fiber was given as 2-3 m. Is it correct? The reviewer thinks that this is in mm, not m?

In Fig.6, applied voltage was not indicated.

Number of significant digits of physical quantities measured was not unified throughout the manuscript. Did the temperature really have five significant digits? The reviewer does not believe that.

The reviewer concluded that the present manuscript is insufficient in quality for publication on the Fluid.

Round 2

Reviewer 2 Report

Authors have revised their manuscript thoroughly and I am satisfied with revision.

Reviewer 3 Report

The reviewer looked over the authors response and the revised manuscript, and no fatal flaw was found in the design of study to reach the conclusion the authors claimed.